# Reliability and Validity of a Novel Reactive Agility Test with Soccer Goalkeeper-Specific Movements

**DOI:** 10.3390/sports10110169

**Published:** 2022-10-31

**Authors:** Yosuke Abe, Hisataka Ambe, Tomoyasu Okuda, Masao Nakayama, Noriteru Morita

**Affiliations:** 1Graduate School of Comprehensive Human Science, University of Tsukuba, Tsukuba 305-8577, Japan; 2Department of Sports Cultural Studies, Hokkaido University of Education, 2-34-1 Midorigaoka, Iwamizawa 068-8642, Japan; 3Faculty of Health and Sport Sciences, University of Tsukuba, Tsukuba 305-8577, Japan

**Keywords:** goalkeeper, change in direction, football, coordination, quickness

## Abstract

The purpose of this study was to develop a reactive agility test with soccer goalkeeper (GK)-specific movements (G-RAT) and to examine the reliability and validity of college-aged GKs. We designed a five-branch star-shaped course with diving and ball-catching movements under reactive conditions. In the setup, a start–goal line was set on the top of a branch and 3.5 m away from the center of the star-shaped setting. Content validity was assessed by six experts, and the item-content validity index (I-CVI) was calculated. Thirty-three male GKs performed the test trial twice. One test trial of G-RAT consists of three shuttles from the start–goal line to diving and ball-catching. For the reactive condition, GKs were instructed on which ball directions should dive when their body trunk reached 1.5 m away from the start–goal line. GKs were classified into regular (R) or non-regular (NR) groups. The intraclass correlation coefficient (ICC) and the receiver operating characteristic (ROC) curve analyses were used to assess the reliability and predictive power as convergent validity. The I-CVI was 0.83, which was greater than the acceptable level of 0.78. The ICC value was 0.94 (*p* < 0.01; 95% confidence interval (95%CI), 0.88–0.97). The GKs completed the test 14.3 ± 0.7 and 15.3 ± 1.0 s in the R and NR group (*p* < 0.01; Cohen’s d = 0.89), respectively. The area under the curves of G-RAT was 0.80 (95%CI, 0.64–0.96). These results show that a GK-specific agility test under reactive conditions would have sufficient reliability and both content and convergent validity in college-aged GKs.

## 1. Introduction

A soccer goalkeeper (GK) is a key defensive player in the most critical position against conceding a goal. While GKs engage in low-intensity movements for most of the duration of a game match, they also need to defend their goal in a critically defensive situation by performing short sprinting and/or diving toward the ball as quickly as possible. Most previous studies for soccer GKs have focused on and examined quick movements (e.g., short straight sprinting and agility), and jump and diving actions. 

A previous review reported the physical and physiological characteristics of GKs based on the comparison with those of field players [1]. According to White et al. [1], GKs had greater performance in jump and diving movements than field players despite similar levels of physical skills, such as sprinting and jumping, and physiological characteristics such as muscle strength, but not aerobic capacity [2,3,4,5,6,7]. In addition, following systematic review search procedures by using Scopus and PubMed databases, we searched previous studies related to sports (i.e., “soccer”, “football”), position (i.e., “GK”, “goalkeeper”, “goalie”), physical abilities (i.e., “fitness”, “physical fitness”, “change of direction”, “agility”, “sprint”, “turn”, “jump”) and game-specific movements of GKs (i.e., “diving”, “save”). We found 45 out of 166 papers that investigated the physiological characteristics of young male GKs. Eleven out of those forty-five papers assessed common agility (e.g., Illinois and 505 agility tests), change in direction performance, and diving movements of GKs. Only three out of those eleven papers focused on agility and diving performances of GKs in simulated practical game situations. There is insufficient evidence on the practical ability-related fitness characteristics of GKs, especially for agility and diving movements in practical situations.

A previous study indicated that elite GKs were faster than non-elite GKs in a short sprint and diving agility test. [8] Although there was no significant difference between the first and substitute GKs in a single diving movement test (single test), the first GK showed greater performance in a complex diving test (diving twice in a test) than substitute GKs. [9] Based on those previous works, GKs needed to improve short sprinting, change in direction abilities, and skillful movements (i.e., diving for the ball and recovery movements after diving); therefore, practical and quantitative assessment of these abilities are required for both coaches and players.

In contrast, according to a review paper, [10] the definition of agility is proposed as “a rapid whole-body movement with change of velocity or direction in response to a stimulus”. In addition, from a practical point of view, GKs are required to reactively respond to opponents attacking players. Considering the definition of agility and practical situations of GKs, agility tests for GKs would need to include reactive conditions for assessing practical agility performance. However, to our knowledge, no study has investigated agility performance and GK-specific movements under reactive conditions.

Thus, this study aimed to develop a new GK-specific agility test with diving movements under reactive conditions and to assess the validity and reliability of the test. The relative and absolute reliability of the test is evaluated by using the test–retest procedure. The validity of the test is determined by content validity, which is based on an expert’s assessment, and convergent validity, which is based on participation in official matches.

## 2. Materials and Methods

### 2.1. Development of Goalkeeper-Specific Reactive Agility Test (G-RAT)

Based on test course settings in previous studies [8,11], the width and length of the goal and goal area, we designed a five-armed star-shaped test course setting for a new GK-specific reactive agility test (G-RAT; Figure 1). When a GK is positioned at the middle of the goal, he/she is required to move in the lateral direction by approximately 3.7 m (4 yards; one-half of the width of the goal) and move in the forward direction by approximately 3–4 m (4 yards; over one-half of the length of the goal area) in most practical situations. 

On the G-RAT settings, a turning point was positioned 3.5 m in front of the start line, and balls are placed on the tops of four branches at a 3.5 m distance from the turning point. Both side branches were set at 90 degrees angle from the start line-to-turning point line, and both diagonal branches were set at a 45 degrees angle from the start line-to-turning point line. A call line for instructing directions was set up at a 1.5 m distance from the start line.

One test trial of G-RAT consisted of three shuttles from the start line to diving and ball catching on branches. Participants performed the test trials as fast as possible a total of two times with at least 3 min rest intervals. The participants were instructed to pass through the turning point during out-bound and in-bound actions. The examiner instructed on which ball to dive to visually observe when the participant’s trunk reached the call line. The participants were briefed beforehand on how to approach the diving motion of a GK that might occur in a real game and were prohibited from sliding from their feet to the ball. When participants were diving for balls during the tests, the GKs needed to fully grab the ball using both hands.

The total patterns for which ball to dive in the G-RAT setting were calculated, and the courses with duplication of same ball diving (e.g., A-B-A) were excluded from test settings. A total of 24 courses were obtained and randomly numbered, and those were used in the test trials.

A 3.5 m short sprint section from a start line to the turning point and 3.5 m diving sections from the turning point to the balls had been set based on a distance of half of the width of the soccer goal. In addition, the number of shuttles for the measurement had been determined by the time required for the completion of the G-RAT. In pilot experiments, college-aged GKs took on average >20 s to complete four shuttles of G-RAT, whereas approximately 15 s were required to complete three shuttles. According to previous studies using the Illinois agility test [12,13], their running times were approximately 15 s; therefore, we chose three shuttles for G-RAT.

### 2.2. Content Validity of the G-RAT

Six experts, four with PhDs and two with more than twenty years of experience as soccer coaches in Japan, evaluated the content validity of the G-RAT. These experts were asked to answer whether the content and measurement setting of the G-RAT was relevant or not to the agility performance of soccer GKs in actual game situations. Prior to the validity assessment of the G-RAT, the experts were provided with the purpose and implication of the study and gave consent to participate in this study. Then, the experts were presented with the definition of agility, [10] settings of the G-RAT, a short video in which GKs performed the G-RAT, and running times of the G-RAT in college-aged GKs (Please see Appendix A). Based on these, the experts assessed the content validity of the G-RAT as an agility test for GKs using the following 4-point Likert scale [14]: score 1 = not relevant, 2 = somewhat relevant, 3 = quite relevant, and 4 = highly relevant. The scores were collected by using Google Forms (Alphabet, Inc., Mountain View, California, United States) with anonymity. A score of 1 or 2, which was an irrelevant rating, and 3 or 4, which was a relevant rating, were converted to dichotomous variables “0” and “1”, respectively. As the main outcome indicator, the item content validity index (I-CVI) was computed as the number of experts who awarded a relevant rating of “1” divided by the total number of experts. As secondary assessments, the experts answered on the degree of validity of the G-RAT setting, especially for the distance from the start-and-goal line to balls via the turning point and the diving movements. The I-CVI for the distance and the diving movement was computed in the same way.

### 2.3. Experimental Participants

A total of thirty-three male GKs competed in the 1st division of University Leagues in the Kanto and Kansai areas during the 2017–2018 seasons.

Participants and coaches were given the purpose and procedures of the present study with documents and verbal explanations. Written informed consent was obtained from all participants. The examiner obtained information about age, height, body weight, history of GK experience, league affiliation, and the number of times the GK participated in official games in the season. Participants were divided into two groups: a regular group (R group) and a non-regular group (NR group). GKs in the R group played or were registered for official games at least once during the season. The GKs in the NR group were not registered for official games in the university leagues. All tests were performed in the afternoon on outdoor soccer fields with artificial grass surfaces. Participants wore soccer gear and boots. Prior to the measurements, participants engaged in a 10 min period of warm-up exercises, which consisted of dynamic stretching, jogging, short sprints at submaximal intensity, and ball drill exercises for GKs. There was no rain, and the air temperature and humidity were 11–21 °C and 40–69%, respectively. This study was conducted in accordance with the Declaration of Helsinki and with the approval of the Research Ethics Committee of Hokkaido University of Education (#2019055003).

### 2.4. Agility and fitness Measurements

Participants performed a total of four fitness tests as follows: (1) 20 m sprint, (2) countermovement jump with arm swing (CMJwA), (3) G-RAT, and (4) simulated G-RAT without diving. We measured 20 m sprint and CMJwA to assess sprinting and jumping abilities of our participants, as these abilities have been suggested to be important physical fitness characteristics for GKs [15]; therefore, these measurements would provide us with information regarding differences in physical fitness. For measuring the 20 m sprint performance, participants ran for a distance of 20 m as fast as possible after being signaled by an examiner. The examiner measured the running time using a handheld stopwatch. For the CMJwA measurement, flight time was measured using a mat switch (Multi Jump Tester, DKH Inc., Tokyo, Japan) connected to a relay circuit and personal computer. CMJwA was calculated using the following equation: CMJwA (m) = (1/8) × 9.81 × (t)^2, where t = flight time (s) [16]. A simulated G-RAT without diving was set up using the same settings as the G-RAT. For the measurement of the G-RAT without diving, four 60 cm tall cones were set, and participants were instructed to touch the cones instead of diving and ball-catching. All tests were performed twice in total. 

### 2.5. Statistical Analysis 

Data are presented as means, standard deviation (SD), and 95% confidence interval (95%CI). Comparisons between R and NR groups were conducted using an independent Student’s *t*-test. All statistical analyses were conducted with α = 0.05, using IBM SPSS statistic, version 23 (IBM Inc., Tokyo, Japan). 

#### 2.5.1. Reliability

The relative reliability of the G-RAT was assessed by using an intraclass correlation coefficient (ICC, model 1,2 [8]) with 1st and 2nd G-RAT trial times. In addition, systematic errors of the tests were evaluated by visual inspection of Bland–Altman analysis. Absolute reliability assessment used standard error of measurement (SEM) and smallest worthwhile change (SWC). The SEM was calculated using the following formula [17,18,19]: SEM = SD × √(1 − ICC). The SWC was calculated using 0.2 × SD. 

#### 2.5.2. Sensitivity

The sensitivity of the test was assessed by comparing the SWC and SEM, where SEM < SWC indicated good sensitivity, SEM similar to SWC was rated satisfactory, and SEM > SWC was rated as marginal sensitivity [20]. The limit of agreement (LOA) was calculated as the mean of the two running times ±1.96 SDs. 

#### 2.5.3. Convergent Validity

The discriminant powers between the R and NR groups were assessed using the area under the curve (AUC) values of a receiver operating characteristic (ROC) analysis of 20 m sprints, CMJwA, height, and G-RAT and simulated G-RAT without diving. Comparing the discriminant powers, we assessed the convergent validity of the variables with the competitive performance of GKs according to the official statistics of game match participation.

## 3. Results

### 3.1. Content Validity of the G-RAT

As an agility test for soccer GKs, five out of six experts assessed a score of 4; thus, the I-CVI of the G-RAT was calculated as 0.83. In secondary assessments, the I-CVI for the distance and diving movements of G-RAT was 0.83 (5/6) and 1.00 (6/6), respectively.

### 3.2. Reliability of G-RAT

Thirty-three GKs participated in and performed the G-RAT test. Their demographic characteristics such as height, body weight, age, and GK experience are shown in Table 1. The value of ICC of G-RAT was 0.94 (95%CI, 0.88–0.97) and it was categorized as “Almost perfect” (Landis and Koch, 1977, [21]). In addition, the sensitivity measurement of G-RAT showed “marginal” due to the SWC being larger than the SEM (Table 2; Faber et al., 2006^,^ [17]). Figure 2 shows Bland–Altman plots of the G-RAT. The dotted lines show the upper and lower LOA. From the visual inspection, there was no systematic error in the G-RAT.

### 3.3. Group Comparison of Physical Fitness

Figure 3 shows test results of the R and NR groups, respectively, regarding height (182.0 ± 6.3 cm vs. 181.3 ± 6.0 cm), weight (78.9 ± 7.4 kg vs. 75.2 ± 7.1 kg), 20 m sprint (3.3 ± 0.2 s vs. 3.4 ± 0.2 s), CMJwA (46.2 ± 7.0 cm vs. 45.8 ± 4.8 cm), simulated G-RAT without diving (11.7 ± 0.6 s vs. 12.1 ± 0.7 s), and G-RAT (14.3 ± 0.7 s vs. 15.3 ± 1.0 s). No significant differences were observed between the R and NR groups in the 20 m sprint, CMJwA, and simulated G-RAT without diving. However, the R group was faster than the NR group in the G-RAT (*p* < 0.05; Cohen’s d = 0.89).

### 3.4. Discriminant Power

Figure 4 shows AUC plots from ROC analyses. AUC values were not significant in 20 m sprint (AUC, 0.59; 95%CI, 0.38–0.80), CMJwA (AUC, 0.53; 95%CI, 0.26–0.79), body height (AUC, 0.45; 95%CI, 0.22–0.68) and simulated G-RAT without diving (AUC, 0.64; 95%CI, 0.43–0.85). However, only the AUC of G-RAT was statistically significant at 0.80 (95%CI, 0.64–0.96, *p* < 0.01). In addition, the cut-off time of G-RAT for game participation was 15.0 s. 

## 4. Discussion

### 4.1. Major Findings

This study aimed to develop a new GK-specific agility test with diving movements under reactive conditions and to evaluate the validity and reliability of the test. The major findings of this study were as follows: (1) the G-RAT was a test with a high level of practically absolute and relative reliability, and (2) G-RAT has a practically enough level of content and convergent validity.

### 4.2. Content Validity of the G-RAT

According to previous studies regarding the content validity index [22,23], an I-CVI greater than 0.78 was acceptable for content validity. The I-CVI of the G-RAT assessed by six experts was 0.83. In addition, the I-CVI for the distance and the diving movement was 0.83 and 1.00, respectively. In practical situations, GKs have to sprint and dive or jump to the ball. Based on the movement construct of the G-RAT and our results for the I-CVIs, it seems that the G-RAT has an acceptable level of content validity as an agility test for soccer GKs.

### 4.3. Reliability of the G-RAT

ICC value indicating the reliability of the G-RAT was 0.94 and showed “almost perfect” as the ideal reliability level is more than 0.85 [21]. ICC values of GK diving tests were 0.73–0.91 by Knoop et al., [9] and 0.90–0.95 by Rebelo-Gonçalves et al. [8] In addition, the tests in the two previous studies by Knoop et al. and Rebelo-Gonçalves et al. did not include recovery movements after diving for the ball. However, GKs in our G-RAT had to recover their position after diving for the ball and make a short sprint to return to the start line. Considering those, it seems that G-RAT would have enough levels of reliability for assessing GK’s skillful movements with diving and recovering. However, the SEM value as absolute reliability index was 0.07 s (0.4%), and the SEM value was larger than the SWC value (0.06 s). According to sensitivity classification by Hopkins, [24] the SEM value is classified as “marginal.” Other previous studies showed that SEM < 5% was sufficient for absolute reliability. [25,26] Taken together, although it remains that the use of G-RAT for detecting the improvement of GK-specific reactive agility should be used with caution in practical settings, the G-RAT has practically high reliability as a GK-specific reactive agility test.

### 4.4. ROC Analysis

ROC analysis was used to calculate cutoff points for screening tests. Within sports science, it has also recently been used to assess fitness performance tests and target value settings [26,27,28]. In the present study, AUCs of body height, CMJwA, and 20 m sprint were not significant and showed no differences between the R and NR groups, suggesting that these measurements would not differentiate GK competitive performance at the collegiate level of GKs. Although the AUC value of G-RAT without diving was 0.64 with no statistical significance, that of G-RAT with diving was 0.80 with statistical significance. As both types of G-RAT times were calculated under reactive conditions and the same course settings, the difference between the discrimination powers of both tests was based on whether diving and recovery movements were present. Thus, levels of reactive agility related to GK-specific movements could contribute at least partially to the opportunity for participation in games. Taking these results into account, it is useful to assess the game participation of GKs based on the proficiency level of diving and recovery movements, which are specific movements of GKs, and encompass agility under reactive conditions. 

Validity has consisted of components such as discriminant, concurrent, and convergent validity. Because GK performance tests of previous studies were targeted to assess different abilities of GKs from G-RAT [8,9], we could not use them as gold standard indicators to assess concurrent validity. Convergent validity is assessed by a relationship between known and new indicators. It appears that the opportunity for game participation of GKs has been determined by the competitive performance of GKs assessed by team coaches; thus, participation and registration in official games could indicate their defensive and competitive performances of GKs. Considering that agility performances assessed by G-RAT could predict GK competitive performance to some extent, it seems that G-RAT would have convergent validity as a reactive agility test related to complex movements for GKs needed for real game matches.

## 5. Limitation and Strength

Although the ICC value of the G-RAT in this study was 0.94, the SEM value was larger than SWC. Taking into account the above, results from the present study have remained. Based on the above levels of absolute reliability of the G-RAT, it appears that improvements are still required. Another limitation is that we used a handheld stopwatch to determine the results, but not electronic timing systems. An original electronic system with measuring software would be required for accurate and individual measurement of each section. Moreover, classification into the R and NR groups in this study was based on the opportunity for game participation determined by their coach’s assessment; therefore, we could not utilize quantitatively standardized GK performance indicators. In addition, although GKs in R and NR groups engaged in a similar training regimen, we did not manage recovery duration following the last intense physical activity such as simulated game practices. If there was a difference between the groups, we would not rule out the possibility that the difference could affect to some extent our results from the ROC analysis and group comparison. As noted in the introduction, although we systematically searched within the literature on physical fitness and dynamic movement of GKs during game matches, we could not find quantitatively standardized GK performance indicators. When such indicators are developed in the future, further studies will be needed to investigate quantitative competitive performance indicators for GKs.

## 6. Conclusions

In conclusion, we developed a new GK-specific agility test, G-RAT, consisting of multiple direction short sprint and GK-specific skillful quick diving movements under reactive conditions, and demonstrated its reliability and both content and convergent validity. From the practical viewpoint, regularly evaluating the agility performance of GK with G-RAT would be useful for grasping the proficiency of diving and recovery movements of GKs.

## Figures and Tables

**Figure 1 sports-10-00169-f001:**
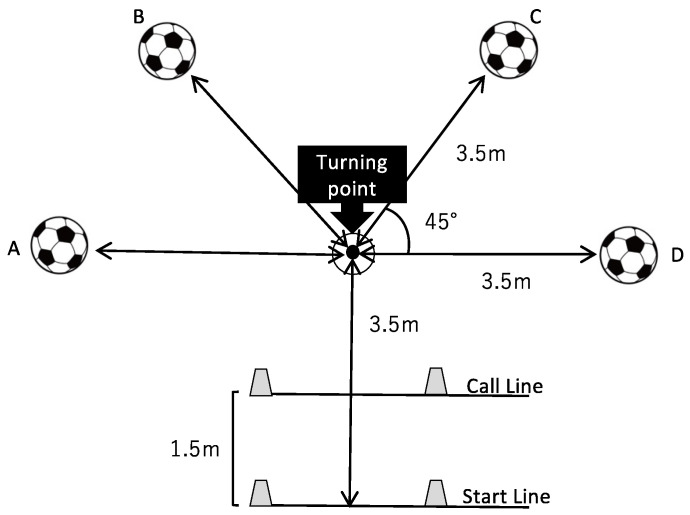
Schematic diagram of test sets of the G-RAT. Start line, the start and goal line for the tests; call line, the line was a marker used for giving verbal or optical instruction; turning point, a reference, and turning marker. The letters A, B, C, and D in the schematic diagram represent direction markers. Participants in the G-RAT measurement receive instructions from the examiner on where to dive while sprinting from the startline to the turning point.

**Figure 2 sports-10-00169-f002:**
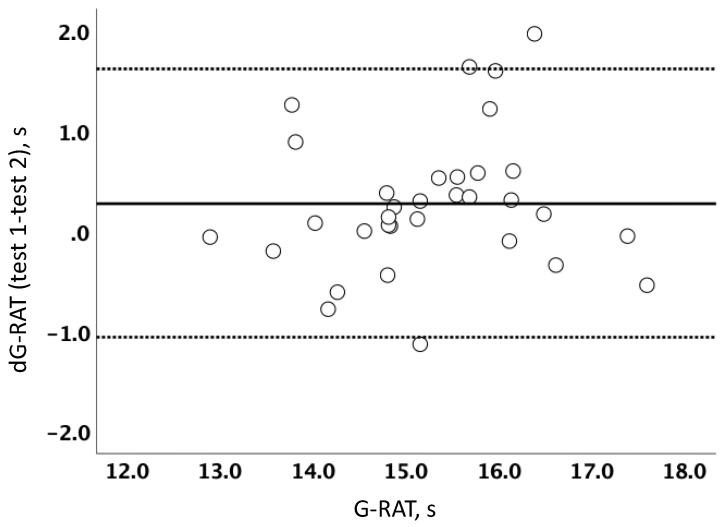
Bland–Altman plots for the total running times of the G-RAT. Solid and dotted lines denote means and lower and upper limits of agreement. G-RAT, goalkeeper-specific reactive agility test; dG-RAT, the difference between test 1 and test 2 in the G-RAT.

**Figure 3 sports-10-00169-f003:**
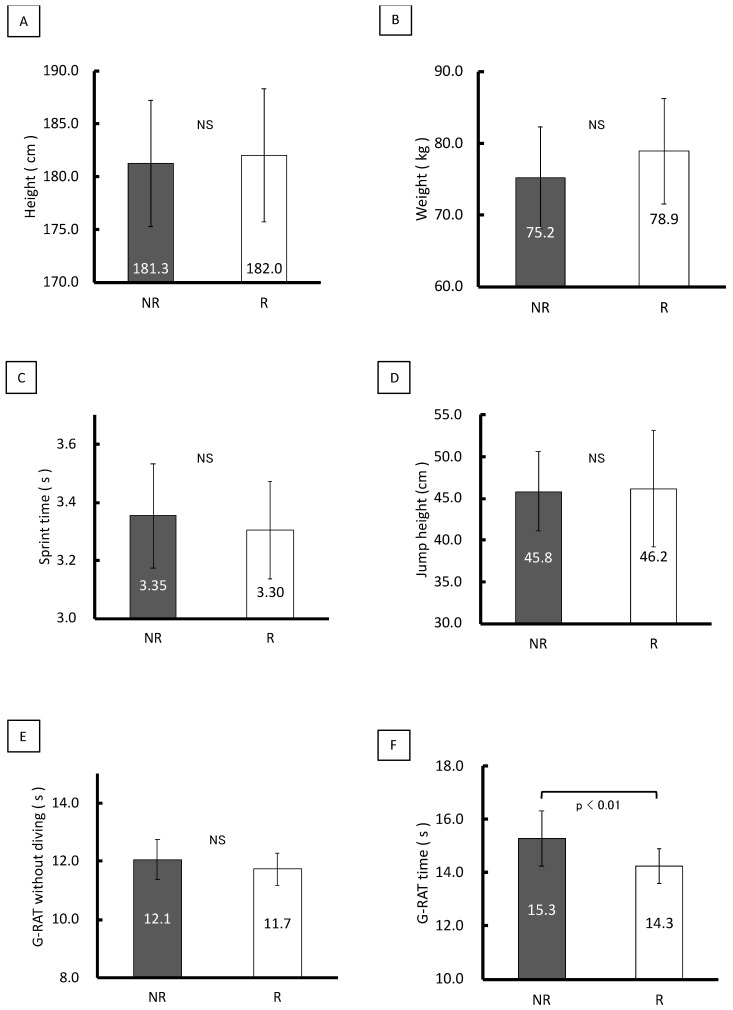
Results of G-RAT, CMJwA, 20 m sprint, and height in the R and NR groups. NR, non-regular (*n* = 47); R, regular (*n* = 22); NS, not significant; CMJwA, countermovement jump with arm swing; G-RAT, goalkeeper-specific reactive agility test. (**A**), height; (**B**), weight; (**C**), 20 m sprint; (**D**), CMJwA; (**E**), G-RAT without diving; (**F**), G-RAT with diving.

**Figure 4 sports-10-00169-f004:**
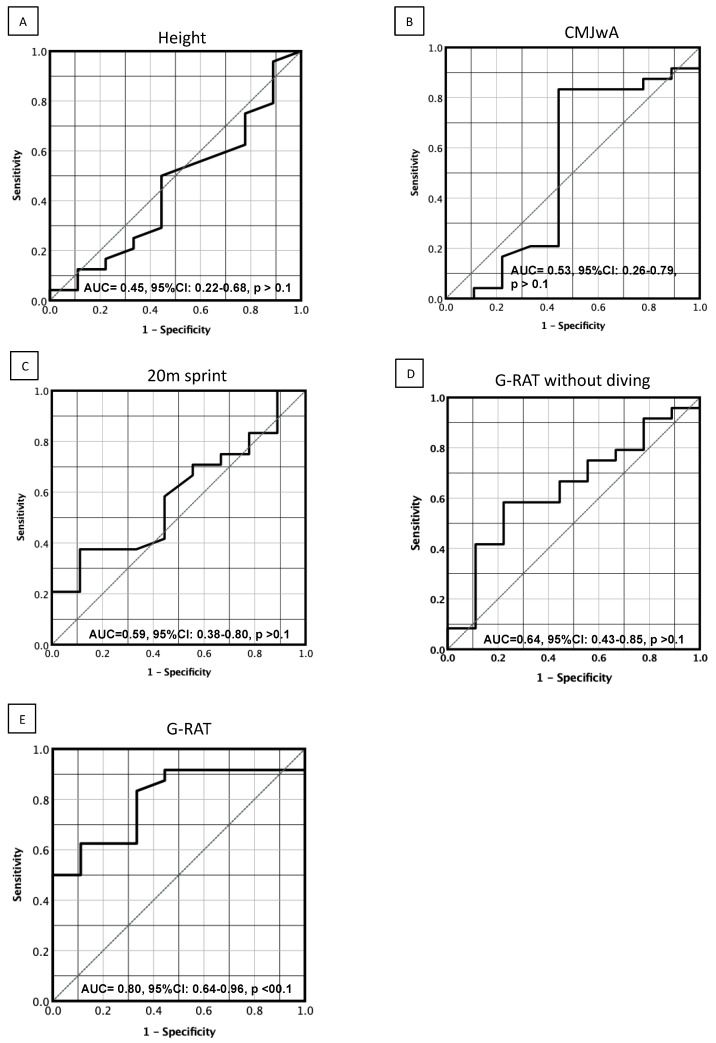
ROC curves for G-RAT, CMJwA, 20 m sprint, and height. CMJwA, countermovement jump with arm swing; RA, reactive agility; RAD, reactive agility and diving; AUC, area under the curve; 95%CI, 95% confidence intervals. (**A**), height; (**B**), CMJwA; (**C**), 20 m sprint; (**D**), G-RAT without diving; (**E**), G-RAT.

**Table 1 sports-10-00169-t001:** Demographic characteristics of participants.

	Total	Regular	Non-Regular
*n*	33	9	24
Height (cm)	181.5 ± 6.1	182.0 ± 6.3	181.3 ± 6.0
Weight (kg)	76.2 ± 7.3	78.9 ± 7.4	75.2 ± 7.1
Age (year)	19.6 ± 0.9	19.6 ± 0.5	19.7 ± 1.1
GK experienced (year)	8.5 ± 3.0	9.0 ± 1.7	8.3 ± 3.4

Values are means ± standard deviations.

**Table 2 sports-10-00169-t002:** Reliability indices of G-RAT.

Variable	Mean T1	Mean T2	Mean	SD	ICC (95% CI)	Mean Difference (T1-T2)	SD Difference	SEM	SWC	LOA Lower/Upper Limits
G-RAT, s	15.4	15.1	15.3	1.1	0.94 (088–0.97)	0.3	0.7	0.07	0.06	–1.05/1.61

T1, test 1; T2, test 2; SD, standard deviation; ICC, intraclass correlation coefficient; 95% CI, 95% confidence intervals; SEM, standard error of measurement; SWC, smallest worthwhile change; LOA, limits of agreement.

## Data Availability

The data presented in this study are available on request from the corresponding author.

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
