# Peer review of "Reliability and Validity of a Novel Reactive Agility Test with Soccer Goalkeeper-Specific Movements"

_sports, 2022, doi:10.3390/sports10110169_

Round 1

Reviewer 1 Report

General comments

The work aims to develop a reactive agility test with soccer goalkeeper (GK)-specific movements (G-RAT) and to examine the reliability, content validity and convergent validity of this new test in college-aged GKs, which is an interesting issue related with player’s performance. The methodology, tools and statistical approach are adequate. The work is clearly presented, and the document is clearly written. However, there are some issues which are outlined below that have to be considered:

Specific comments

 Comment 1  

Abstract: they should start with a first paragraph describing the background.

Comment 2

Introduction section: The Introduction is very short. The rationale behind the study was not given properly in the introduction.

Comment 3

References used in the introduction are relatively old. Authors use only 4 references in the introduction!! Please add more references related to the subject of the study.

 Comment 4

Methods section: More information should be provided about the participants’ characteristics. Moreover, authors should describe an anthropometric measurements protocol.

Comment5

Methods section: More information should be provided about test conditions: outdoor/indoor, the timetable for taking tests, ambient temperature, humidity, sports clothing required for subjects…

 Comment 6

A video of the new G-RAT test should be proposed as supplementary data.

Comment 7

The manuscript is completely missing the practical implications of the study! Please do it…

Comment 8

Based on the analysed variables, how the authors intend to use their findings?

Author Response

Dear, Reviewer #1

We deeply appreciate your pertinent comments and suggestions. Following your comments, we revised our manuscript. I’m sorry to bother you again, please review our revised manuscript. We will be happy to get valuable comments.

Comment 1

→Although we would like to add the background of the present study, we could not add a sentence due to the word limit of the Abstract (up to 250 words).

Comment 2

→We revised the introduction on some points. The revised introduction may be still shorter than common papers, but as the reviewer knows, previous studies for GKs are not many. We consider that the minimum elements of rationales necessary for our research question would be described. If the reviewer recommends again adding any specific issue for our introduction, could you please inform us of that then we will be happy to add that.

Comment 3

→LINE 39-41: We added six references in the introduction as follows “According to White et al.,1 GKs had greater performance in jump and diving movements than field-players despite similar levels of physiological abilities, such as sprinting, jumping and muscle strength abilities, but not aerobic capacity (Arnason et al. 2004; Gil et al. 2007; Nikolaidis et al. 2014; Towlson et al. 2017; Taşkin 2008; Aziz et al. 2008)”.

Comment 4

→Anthropometric characteristics were obtained from self-reports.

Comment 5

→LINE 143-146: We added the following sentences “All tests were performed on outdoor soccer fields with artificial grass surface in the afternoon. Participants wore soccer wear and boots for goalkeepers. Measurement days were not rain, and air temperature and humidity were 11- 21ºC and 40- 69%, respectively.”

Comment 6

→We added a video that included a scene of G-RAT measurement.

Comment 7.8

→We pointed out the implication of the G-RAT on lines 305-307 in the conclusion section. If the reviewer recommends again adding more sentences for practical implications, could you please inform us, then we will be happy to add that.

We think our manuscript clearly improved by the reviewer’s comments. Thank you very much for your constructive suggestions.

Author Response

Dear Reviewer #2,

We deeply appreciate your pertinent comments and suggestions. Following your comments, we revised our manuscript. I’m sorry to bother you again, please review our revised manuscript. We will be happy to get valuable comments.

Note: we indicate line numbers of the revised manuscript in the parenthesis.

LINE 32: We revised following your comments.

LINE 38 (line 39 in the revised MS): We revised the point following your suggestions.

LINE 39 (lines 39-41 in the revised MS): We modified the sentence “despite similar levels of physiological abilities, such as sprinting, jumping and muscle strength abilities, but not aerobic capacity and add six references.”

LINES 40-45 (lines 43-45 in the revised MS): We added the sentences “sports (i.e., `soccer’, `football’), position (i.e., ‘GK’, `goalkeeper’, goalie), physiological characteristics (i.e., `fitness’, `physical fitness’, `change of direction’, `agility’, `sprint’, `turn’, `jump’) and game-specific movements of GKs (i.e., `diving’, `save’).”

LINES 53-55 (lines 82-85 in the revised MS): We modified the sentence “GKs have needed to improve short sprinting, change of direction abilities and skillful movements (i.e., diving to the ball and recovering movements after diving).”

LINE 72 (line 103 in the revised MS): We revised

LINE 73 (line 104 in the revised MS): We revised.

LINE 79 (line 83 in the revised MS): We revised.

LINE 85 (line 89 in the revised MS): We deleted “is.”

LINE 87 (line 91 in the revised MS): Rephrased that “The participants were instructed to pass through the turning point in out-bound and in-bound actions.”

LINE 88 (line 93 in the revised MS): We changed following your suggestions.

LINE 89 (line 93 in the revised MS): We deleted it.

LINES 99 and 100 (line 103 in the revised MS): We revised.

LINES 102-103 (lines 105-107 in the revised MS): We revised following your comments.

LINE 104 (line 108 in the revised MS): We rephrased from “their required times” to “their running times”.

LINE 128 (line 132 in the revised MS): We revised the point following your suggestions.

LINES 129-131: We deleted the sentences.

LINE 137 (line 139 in the revised MS): We changed to “number of times the GK participated.”

LINE 138 (lines 140-143 in the revised MS): We modified the sentences from “Participants were divided into two groups: a regular group (R group) based on participation and registration of official games at least once or a non-regular group (NR group) without participation and registration of official games in the university leagues.” to “Participants were divided into two groups: a regular group (R group) or a non-regular group (NR group). GK of the R group had experiences of playing or registration for the official games at least once during the seasons. The NR group had no experiences of registration of the official games in the university leagues.”

LINES 149 (line 154 in the revised MS): We added a study by Condello et al.(2013).

LINES 150-152 (line 155 in the revised MS): We modified the sentences from “Participants commenced with the signal from an examiner when measuring a 20-m sprint and were measured the running time from the start line until the goal by a handheld stopwatch.” to “For measurement of 20-m sprint performance, participants ran 20 m as fast as possible after the signal from an examiner. the examiner measured their running time with a handheld stopwatch.”

LINES 153 (lines 158-161 in the revised MS): We revised the sentences, but we couldn’t cite the article by Ishida et al., because the method in the paper was different from our study used. We added the following sentence: “CMJwA was calculated by the following equation: CMJwA (m)= (1/8)* 9.81* (t)^2, t= flight time (s).”

LINE 156 (line 158 in the revised MS): We revised the point following your suggestions.

LINE 161(line 168 in the revised MS): We revised the point following your suggestions.

LINE 166(line 172 in the revised MS): We added a reference regarding this analysis.

LINE 202(lines 209-210 in the revised MS): We added data for height and weight.

Figure 3: We revised the unit abbreviation.

LINE 243 (lines 250-255 in the revised MS): We added the following sentence: “In addition, the tests in the two previous studies by Knoop et al. and Rebelo-Gonçalves et al. did not include recovery movements after diving the ball. However, GKs in our G-RAT had to recover their position after diving for the ball and make a short sprint to return to the start line. Considering those, it seems that G-RAT would have enough levels of reliability for assessing GK's skillful movements with diving and recovering.”

LINE 270: We deleted the sentences. Thank you for your suggestion.

LINE 284 (lines 292-293 in the revised MS): We modified the sentence from “Taking into account the above, results from the present study have remained to be improved.” to “Based on the above levels of absolute reliability of the G-RAT, it seems that G-RAT still remains to be improved.”

We think our manuscript clearly improved by your constructive comments. Thank you very much for your constructive suggestions.

Round 2

Reviewer 1 Report

The manuscript is improved. All my concerns were addressed.

Author Response

Dear Reviewer #1,

We deeply appreciate your pertinent comments and suggestions. 

Reviewer 2 Report

The authors addressed the requests made during the first round of reviews and the manuscript is suitable for publication.

Only minor a spell check needs to be made:

LINE 210: “(82.0±6.3 cm” please correct height to “182.0

LINE 159: please include a reference for the use of countermovement jump with arm swing in football (i.e. Trecroci et al., 2020 https://doi.org/10.3390/sports8080108)

Author Response

Dear Reviewer #2,

We deeply appreciate your pertinent comments and suggestions. Following your comments, we revised our manuscript. I’m sorry to bother you again, please review our revised manuscript. 

LINE 210: “(82.0±6.3 cm” please correct height to “182.0

Reply: We added “1.”

LINE 159: please include a reference for the use of countermovement jump with arm swing in football (i.e. Trecroci et al., 2020 https://doi.org/10.3390/sports8080108)

Reply* We added the reference.

Thank you for your support.